# Study protocol: The effect of a low-carbohydrate enteral nutrition formula on postoperative hyperglycemia in non-diabetic patients with esophageal cancer: A randomized exploratory phase II trial (ENLICHE study)

**Masayoshi Terayama[1], Yu Imamura[1]\*, Toru Kitazawa[2], Naoki Miyazaki[3], Misuzu Ishii[4], Kumi Takagi[4], Kengo Kuriyama[1], Naoki Takahashi[1], Masahiro Tamura[1], Akihiko Okamura[1], Jun Kanamori[1], Masayuki Watanabe[1]**

**1** Department of Gastroenterological Surgery, Cancer Institute Hospital, Japanese Foundation for Cancer Research, Tokyo, Japan, **2** Department of Diabetology, Endocrinology and Metabolism, Cancer Institute Hospital, Japanese Foundation for Cancer Research, Tokyo, Japan, **3** Department of Clinical Trial Planning and Management, Cancer Institute Hospital of Japanese Foundation for Cancer Research, Tokyo, Japan, **4** Department of Nutritional management, Cancer Institute Hospital, Japanese Foundation for Cancer Research, Tokyo, Japan

\* yu.imamura@jfcr.or.jp

**Data availability statement:** No datasets were generated or analysed during the current study.

## Abstract

### Background

Postoperative hyperglycemia in diabetic patients is a widely known risk factor for postoperative infectious complications (PICs) after esophagectomy; however, the significance of glycemic control in non-diabetic patients is less clear. In diabetic patients, early postoperative management of esophagectomy favors low-carbohydrate enteral nutrition (EN) over standard EN to suppress the risk of glycemic spike. Our single-center, randomized phase II trial seeks to test the hypothesis that low-carbohydrate EN can suppress hyperglycemia in non-diabetic patients who undergo esophagectomy. Herewith we present the study protocol.

### Methods

A total of 50 patients will be enrolled and randomly assigned (1:1 ratio) to standard or low-carbohydrate EN. Randomization will be stratified by operation time (≥560 vs. <560 min) and HbA1c (6.0–6.4% vs. <6.0%). Both EN formula will be fed according to the following protocol: 400 mL/24 h on postoperative day (POD)-1; 800 mL on POD-2; 1200 mL on POD-3 and 1600 mL from POD-4 to POD-8. On POD-9, oral food intake will be initiated. A continuous glucose monitoring (CGM) device will be used to monitor blood glucose levels from POD-1 to −8. The primary outcome is the mean time-in-range (TIR) across the 48 h from POD-1 to −2. TIR is defined as the

All relevant data from this study will be made available upon study completion.

**Funding:** This study is supported by the R&D fund from the Center for Development of Advanced Cancer Therapy, the Cancer Institute Hospital of Japanese Foundation for Cancer Research (CDACT24-13; for MW), and a grant of the 7th Japanese Association for Thoracic Surgery award for young investigators in 2024 (MT). The funders had no role in study design, data collection and analysis, decision to publish, or preparation of the manuscript.

**Competing interests:** The authors have declared that no competing interests exist.

percentage-time that blood glucose remains within the targeted range of 70−180 mg/dL. The primary analysis will calculate the least squares mean difference in TIR over the 48 h (POD-1 to −2) between the two groups, with p-values calculated to test the null hypothesis that the mean difference between the groups is zero. The secondary outcomes will be as follows: 1) the incidence of PICs and/or other adverse events within 30 days after esophagectomy or during the hospital stay; 2) the number of cases requiring any dose alteration in EN formula during monitoring; 3) the number of cases requiring interventions for hyperglycemia or hypoglycemia; 4) the rates in change of nutritional indicators, such as serum albumin, prealbumin, and total protein levels, during the post-surgical hospital stay (vs. those values on the day of admission); and 5) the following CGM indices in relation to the incidence rate of PICs within 30 days after esophagectomy: the mean values for time-above-range (TAR), area under the curve (AUC), and TIR for each POD or from POD-1 to −8. TAR is defined as the percentage of time of a patient is recorded as having hyperglycemia (>blood glucose level of 180 mg/dL), and is indicative of the frequency and duration of hyperglycemia. AUC, which identifies periods of hyperglycemia and provides a comprehensive picture of glucose variability and control in diabetes management, is defined as the area under the curve over blood glucose level of 180 mg/dL on CGM monitoring.

## Discussion

This study is the first to investigate the impact of a low-carbohydrate EN formula on hyperglycemic control during perioperative nutritional management of esophageal cancer. These results will help to outline whether glycemic control should be also considered for non-diabetic patients during hospital care.

## Trial registration

This trial has been registered in the Japanese Registry of Clinical Trials (jRCTs031240081).

## Introduction

Esophageal cancer is the sixth-leading cause of cancer-related death worldwide [1]. Despite significant progress in multimodal therapies, esophagectomy remains the primary curative treatment for esophageal cancer [2,3]. However, esophagectomy is associated with a high incidence of postoperative infectious complications (PICs) and have a significantly negative impact on survival [4,5].

Type 2 diabetes mellitus (DM) is a well-known risk factor for PICs in patients who undergo esophagectomy [6–8]. Enteral nutrition (EN) is widely administered via a gastrointestinal fistula during early postoperative care after esophagectomy [9], with diabetic patients typically fed a low-carbohydrate meal to suppress the chance of a glycemic spike [10,11]. Accumulating evidence suggests that surgical stress

increases insulin resistance through the activation of inflammatory cytokines, leading to hyperglycemia, and this is purported to occur irrespective of diabetic status [12]. Despite these findings, little attention has been paid to the management of postoperative hyperglycemia in non-diabetic patients and, at present, it is unclear whether low-carbohydrate EN would similarly aid in suppressing hyperglycemia in non-diabetic patients.

In a recent, retrospective study in non-diabetic patients after esophagectomy, we found that early postoperative hyperglycemia from postoperative day (POD)-1 to −4 in the intensive care unit (ICU) was associated with a high incidence of PICs and poorer survival [13]. Thus, we seek to test our hypothesis that low-carbohydrate EN can suppress hyperglycemia after esophageal surgery in non-diabetic patients through a randomized, exploratory, phase II trial in non-diabetic patients with esophageal cancer (ENLICHE study). This study will be the first to monitor postoperative blood glucose levels using a continuous glucose monitoring (CGM) device after gastrointestinal surgery. Herewith, we present the study design.

## Materials and methods

### Study design

This is a prospective, single-center, open-label, randomized controlled study to evaluate the efficacy of a low-carbohydrate EN formula in controlling postoperative hyperglycemia in non-diabetic patients with esophageal cancer. This trial is registered with the Japanese Registry of Clinical Trials (jRCTs031240081). Recruitment is ongoing from 14/June/2024 and will continue until 50 patients have been included, approximately 30/November/2025.

### Inclusion criteria

Patients who meet the following inclusion criteria will be eligible for this study: 1) histologically confirmed esophageal cancer, 2) no history of diabetes mellitus or HbA1c < 6.5% on preoperative blood tests, 3) surgical planned procedure is for subtotal esophagectomy with reconstruction using a gastric conduit, 4) aged 20 years or older, and 5) has adequate organ function to tolerate general anesthesia. In accordance with standard preoperative management for esophagectomy in non-diabetic patients, methylprednisolone 250 mg/body will be administered to all patients just before surgery to control the excessive biological response to surgical trauma.

### Exclusion criteria

Patients will be excluded if any of the following are true: 1) they present with a distant metastasis; 2) have a surgical plan for one of the following procedures: palliative esophagectomy, pharyngo-laryngo-esophagectomy, two-staged esophagectomy, or esophagectomy with thoracic duct resection; 3) have received or will need to receive irradiation before surgery, and 4) have an implanted pacemaker with a reported potential for causing malfunctions in blood glucose sensors.

### Registration and randomization

The following data will be preoperatively evaluated for study enrollment: age, sex, birth date, height, weight, blood pressure, heart rate, past medical history, comorbidities, allergies, oral medications, symptoms, Eastern Cooperative Oncology Group Performance Status (ECOG-PS), blood tests, including diabetic status, nutritional, lipid and electrolyte profiles, urine tests, electrocardiogram, chest X-ray, and computed tomography. After obtaining these baseline demographic and clinical data, the investigators will register the patients who meet all of the inclusion criteria. The written informed consent was obtained from all the registered patients.

Eligible patients will be then randomly assigned in a 1:1 ratio to either conventional EN (control group) or low-carbohydrate EN (intervention group, Fig 1). Randomization will be conducted using an electronic data capture system (Viedoc Technologies; Pharma Consulting Group, Uppsala, Sweden), with the assignment sequence generated and concealed within the system. Randomization will be stratified by operation time (≥560 vs. < 560 min), as this factor was

| | STUDY PERIOD | | | | | | | Close-out | |
| | Enrolment | Allocation | Post-allocation | | | | | | |
| | | | ICU | | | General ward | | | |
| TIMEPOINT | −28〜−1 | 0 | 1 | 2 | 3 | 4〜8 | 9 | discharge | 30 days after the final dose |
| **ENROLMENT:** | | | | | | | | | |
| Eligibility screen | X | | | | | | | | |
| Informed consent | X | | | | | | | | |
| Allocation | | X | | | | | | | |
| **INTERVENTIONS** | | | | | | | | | |
| Meiji Main | | | X | X | X | X | | | |
| Glucerna-REX ® | | | X | X | X | X | | | |
| CGM | | X | X | X | X | X | X | | |
| intermittent blood glucose measurement | | | X | X | X | X | | | |
| dosage of enteral nutrition formula (mL) | | | 400 | 800 | 1200 | 1600 | | | |
| **ASSESSMENTS:** | | | | | | | | | |
| Infectious complications | | | X | X | X | X | X | X | X |
| Other complications | | | X | X | X | X | X | X | X |
| Adverse effects | | | X | X | X | X | X | X | X |
| survival | | X | X | X | X | X | X | X | X |

**Fig 1. Flow diagram of randomization of this study.**

significantly associated with the development of PICs in nondiabetic patients in our previous study [13]. Stratification will also consider preoperative HbA1c levels (clinical pre-diabetic range of 6.0–6.4% vs. normal/near-normal range of <6.0%).

## Data management

The collected data will be managed using an Electronic Data Capture system and monitored by an independent Data Monitoring Committee composed of external members free from conflicts of interest. The final handling of each dataset will be determined through discussions between the principal investigator, the research office, the lead statistician, and the data center. Study results will be submitted to an international peer-reviewed journal. If challenges such as delays in patient recruitment, frequent protocol deviations, or an unfavorable risk-benefit ratio arise, the principal investigator will report these to the clinical research board. Should these challenges compromise the feasibility of completing the study or necessitate study revisions, the termination procedure will be initiated. Upon completion of the study, all personally identifiable information will be removed from the dataset. The anonymized data will be securely stored by a designated data manager for a minimum of five years in accordance with data protection regulations.

## Study treatment

**Protocol of enteral nutrition formula.** MEIN (Meiji Co., Ltd., Tokyo, Japan), a standard EN formula with no specific nutrient restriction, will be assigned to patients in the control group, whereas Glucerna-REX (Abbot, Alameda, CA, USA), a low-carbohydrate EN formula, will be fed to those in the intervention group. The ingredients are listed in Table 1. Both formulae will be fed according to the following regimen: 400 mL per 24 h on POD-1; 800 mL on POD-2; 1200 mL on POD-3 and 1600 mL from POD-4 to POD-8. On POD-9, oral food intake will be initiated, with EN switched simultaneously to 1000 mL per 24 h of Hinex Igel, a concentrated liquid food.

In the event of adverse events related to the administration of nutritional formulations, it may become necessary to switch both the control and intervention groups to alternative formulations or reduce the dosage outside the protocol-specified treatment. When Grade 3 or higher adverse events occur (as defined by CTCAE version 5.0), participants in both groups may either transition to a non-protocol nutritional formulation deemed appropriate by the principal or sub-investigator, or reduce the dosage by 50% or more compared to the previous day, while continuing the protocol-specified formulation. When any health issues arise in participant of this study, the principal investigator will ensure appropriate medical care and take any other necessary measures. To mitigate risks associated with unforeseen health issues, clinical research insurance will be secured in advance.

**Glucose monitoring.** The FreeStyle Libre Pro (Abbot Diabetes Care Inc., Alameda, CA) CGM device will be used to monitor glucose changes [14]. A removable sensor will be attached to the patient's outer upper arm upon admission to the ICU just after surgery (POD-0). Blood glucose levels will be continuously monitored from postoperative day (POD) 0 to POD 9. However, due to potential instability in measurements immediately following sensor application, data collected from POD 1 to POD 8 will be used for analysis. Intervention for hyperglycemia or hypoglycemia will be based on our

**Table 1. Ingredients of enteral nutrition formula used in this study.**

| Variable | MEIN (Meiji), standard EN | Glucerna-REX (Abbott), low-carbohydrate EN |
|---|---|---|
| Capacity | 100mL | 100mL |
| Energy | 100Kcal | 100Kcal |
| Protein | 5g | 4.2g |
| Lipids | 2.8g | 5.6g |
| Carbohydrates | 15g | 9.7g |
| Glucidics | 13.2g | 8.8g |
| Dietary fiber | 1.8g | 0.9 |
| Fluids | 84.1g | 85g |
| Vitamin A | 150µg | 104µg |
| Vitamin B1 | 0.25mg | 0.12mg |
| Vitamin B2 | 0.3mg | 0.18mg |
| Vitamin B12 | 0.6µg | 0.3µg |
| Vitamin D | 0.75µg | 0.9µg |
| Folic acid | 50µg | 20µg |
| Vitamin C | 50mg | 11mg |
| Sodium | 80mg | 94mg |
| Potassium | 120mg | 100mg |
| Calcium | 100mg | 70mg |
| Osmotic pressure | 640mOsm/L | 560mOsm/L |

EN, enteral nutrition

conventional monitoring of arterial blood gas tests taken every 6 h during ICU stay from POD-1 to −4, according to the following sliding scale: intravenous administration of 2 units of regular insulin for measurements of 201−249 mg/dL, 4 units for 250−299 mg/dL, 6 units for 300−349 mg/dL, 8 units for 350−399 mg/dL, 10 units for 400 mg/dL or above, and 20 ml of 50% glucose for below 69 mg/dL. However, after surgical trauma, blood glucose levels between 201 and 299 mg/dL are considered as only a "mild increase"; in such cases, insulin intervention will not be required from POD-0 to POD-2.

**Data collection.** For this study, laboratory data, clinical symptoms, and physical examinations will be re-evaluated on POD-14, the day of discharge, and once again after discharge between POD-30 and −44. Any adverse effects related to EN or the use of the CGM sensor will be also monitored.

**Outcomes.** The primary outcome will be the mean time-in-range (TIR) from POD-1 to −2, as measured by CGM. TIR is defined as the percentage of time that blood glucose is within the targeted range of 70–180 mg/dL over a 24-h CGM monitoring period [15–17]. This is an important metric in DM management because it provides a comprehensive view of blood glucose control over time, beyond what HbA1c or single blood glucose measurements can offer.

The secondary outcomes will be as follows: 1) the incidence of PICs and/or other adverse events within 30 days after esophagectomy or during the hospital stay; 2) the number of cases requiring any dose alteration in EN formula during the hospital stay; 3) the number of cases requiring interventions for hyperglycemia or hypoglycemia; 4) the rates in change of nutritional indicators, such as serum albumin, prealbumin, and total protein levels, during the post-surgical hospital stay (vs. those values on the day of admission); and 5) the following CGM indices in relation to the incidence rate of PICs within 30 days after esophagectomy: the mean values for time-above-range (TAR), area under the curve (AUC), and TIR for each POD or from POD-1 to −8. TAR is defined as the percentage of time of a patient is recorded as having hyperglycemia (>blood glucose level of 180 mg/dL), and is indicative of the frequency and duration of hyperglycemia. Higher TAR values thus represent a risk of diabetes-related complications if not managed properly as well as an important goal in diabetes management to maintain optimal glucose control [15]. AUC, which identifies periods of hyperglycemia and provides a comprehensive picture of glucose variability and control in diabetes management, is defined as the area under the curve over blood glucose level of 180 mg/dL on CGM monitoring [15].

**Recruitment.** In clinical trials that examine antidiabetic drugs for diabetic patients, a commonly used efficacy indicator is a 0.5%-reduction in HbA1c or more. Such a change corresponds to a 10% change in TIR. For the present study on non-diabetic patients, a >5% increase in TIR in the intervention group will be set as the minimum clinically significant difference in the mean between the two groups. With a sample size of 48 cases and assuming a standard deviation of 8.5% for the difference in TIR between the groups, the mean difference in TIR between the two groups can be estimated with a confidence interval width of ±5%. Considering potential dropouts, the sample size for this study was set to 25 cases in each group, totaling 50 cases. Approximately 120 esophagectomies are performed each year at our institution. Among these, 12% (n = 14) are performed in patients with diabetes mellitus, indicating an estimated 106 non-diabetic cases annually, based on our previous study [13]. Given that approximately half of the non-diabetic patients are expected to be enrolled in this study, the enrollment of 50 cases is projected to be completed within 1.5 years.

This study protocol was approved by the certified review board of Cancer Institute Hospital (approval number: GKC-2401).

## Statistical analysis

Efficacy analyses will be conducted using the Full Analysis Set (FAS), whereas safety analyses will include all cases based on the intention-to-treat principle. The FAS is defined as the population from which cases are excluded if they have major violations of inclusion or exclusion criteria, did not receive the protocol treatment, have no efficacy data, or withdrew consent during the study and refused the use of all data. In the event that any cases require changes to the enteral nutrition formula or dosage reduction, a subgroup analysis will be conducted comparing groups that followed the protocol according to the original assignment (Per Protocol Set), excluding those cases. In principle, missing data will not be

imputed. However, if necessary, sensitivity analyses incorporating imputation of missing data will be conducted. Specifically, if missing data occur during data collection, sensitivity analyses will be performed on cases in which more than 70% of the planned measurements have been obtained.

The primary analysis will calculate the least squares mean difference, 95% confidence interval, and standard deviation of TIR across the 48 h (POD-1 to −2) between the two groups. The p-value will be calculated to test the null hypothesis that the mean difference between the two groups is zero. For secondary analyses, the mean difference between groups, 95% confidence interval, and standard deviation will be calculated, and group comparisons will be performed using chi-square tests or t-tests to obtain p-values. Given the exploratory nature of this study and the limited sample size, adjustments for multiplicity in the analysis of secondary outcomes will not be performed.

All statistical analyses will be performed using EZR (Saitama Medical Center, Jichi Medical University, Saitama, Japan), a graphical user interface for R (The R Foundation for Statistical Computing, Vienna, Austria) [18].

## Discussion

Esophagectomy is recognized as a highly invasive surgical procedure associated with the highest incidence of PICs among gastrointestinal surgeries [5,19]. Inflammatory cytokines induced by surgical stress are considered significant causative factors for PICs, including pneumonia and anastomotic leakage. Notably, hyperglycemia impairs neutrophil chemotaxis and phagocytosis, thus positioning it as a major risk factor for PICs. Because PICs negatively impact the long-term outcomes of patients with esophageal cancer [2,12,20–25], mitigating PICs represents a crucial clinical objective in the management of esophageal cancer surgery.

Among non-diabetic patients, however, the clinical impact of hyperglycemia is unclear. We recently showed that early postoperative hyperglycemia on POD-1 to −4 in non-diabetic esophageal cancer patients significantly increased the incidence of PICs. These findings highlighted the critical importance of perioperative glycemic control even in non-diabetic patients [13]. Nutritional management is one of the key factors contributing to postoperative hyperglycemia. Patients receive EN using semi-elemental formulas immediately after esophagectomy until they can achieve sufficient oral intake. Glucerna-REX, a low-carbohydrate enteral formula designed with a low carbohydrate energy ratio of 25% and an increased lipid ratio of 50%, is being used for specific patients [26,27], while conventional enteral formulas, such as MEIN, have a carbohydrate energy ratio of approximately 60% and are used more generally post-surgically.

To date, the need for glycemic control in non-diabetic patients has been overlooked, and no studies have directly compared the effects of standard EN with those of low-carbohydrate EN on perioperative blood glucose. Indeed, we surmise that, in situations where postoperative blood glucose management proves difficult with conventional EN like MEIN, Glucerna-REX could be more effective in suppressing blood glucose elevation. This study is the first to investigate the impact the effect of Glucerna-REX on hyperglycemia control during the perioperative nutritional management of esophageal cancer in non-diabetic patients, with a specific emphasis on the importance of glycemic control. Furthermore, no previous studies have comprehensively assessed 24-h blood glucose variability after esophageal cancer surgery. Data related to potential hyperglycemia and hypoglycemia obtained through CGM measurements not only provide essential information for future perioperative management research, but also aid in the development of evidence-based perioperative surgical management strategies. In our previous study, postoperative hyperglycemia, particularly during the early postoperative period (POD 1–2), was associated with poor overall survival in non-diabetic patients with esophageal cancer who underwent esophagectomy [13]. Based on those findings, we plan to explore the potential prognostic benefit of the use of low-carbohydrate enteral nutrition by long-term follow-up in the present study. Depending on the results, a future phase III trial may be designed to examine not only the reduction of PICs, but also the potential survival benefits of low-carbohydrate enteral nutrition.

This study design has several limitations. Firstly, this study does not control potential influences of patients' underlying metabolic disorders or dietary habits on postoperative glucose levels. However, in non-diabetic patients, there is no clear

evidence indicating whether these factors significantly affect early postoperative glucose metabolism. In this study, we plan to conduct subgroup analyses based on metabolic and nutritional factors including triglycerides and LDL cholesterol to provide insights that may contribute to future research development. Secondly, there is a potential limitation associated with instability in glucose measurements during the initial 8–12 hours following sensor insertion, which is performed in the immediate postoperative period. To minimize such variability, data collection begins on the day after surgery, allowing at least 12 hours to elapse after sensor placement. Thirdly, the potential influence of differences in perioperative medications on glucose measurements cannot be completely eliminated. However, this study includes only non-diabetic patients, and individuals taking oral medications that directly affect blood glucose levels are excluded. In our clinical practice, medications not essential for life support—such as antiplatelet and antiepileptic agents—are generally resumed from postoperative day 3. Moreover, any newly initiated medications in the postoperative period are standardized across all patients. Therefore, we believe that the primary endpoint, assessed based on blood glucose levels up to postoperative day 2, is unlikely to be significantly affected. Finally, as this is a pilot study with a small sample size conducted at a single institution, the generalizability of the findings may be limited. In addition, the open-label design may introduce potential bias. If the efficacy and safety of a low-carbohydrate nutritional formula in non-diabetic patients are confirmed, the findings will serve as confirmatory evidence to justify a double-blind, multi-center phase III trial aimed at further evaluating the impact of low-carbohydrate enteral nutrition on reducing PICs after esophagectomy. Furthermore, such a trial should also investigate the potential effects of low-carbohydrate enteral nutrition on lipid metabolism and gut microbiota. In addition, a reduction in infectious complications would demonstrate the cost-effectiveness of low-carbohydrate enteral nutrition, supporting its broader adoption in perioperative management—not only for non-diabetic patients undergoing esophagectomy, but also across a wide range of surgical procedures. Given that the cost of low-carbohydrate enteral nutrition is not significantly different from that of conventional formulas, we believe that, if this study confirms its effectiveness in suppressing early postoperative hyperglycemia, the medical environment should be adapted to facilitate its wider implementation.

## Supporting information

**S1 Study Protocol.** **The study protcol of ENLICHE study translated into English.**
(DOCX)

**S2 Study Protocol.** **The original study protcol of ENLICHE study in Japanese.**
(DOCX)

**S1 Checklist.** **The desciption of SPIRIT 2013 checklist.**
(DOCX)

## Acknowledgments

The authors express our sincere application to Satomi Ando, Mayumi Miyamoto, Rumiko Ohno, Yoshiko Matsui, Sachiko Sawada, Hiroshi Asai, Hidenobu Ishizaki, and Ikumi Haraguchi for their help in the preparation and execution of this study.

## Author contributions

**Conceptualization:** Masayoshi Terayama, Yu Imamura, Toru Kitazawa, Masayuki Watanabe.

**Data curation:** Masayoshi Terayama, Naoki Miyazaki.

**Formal analysis:** Masayoshi Terayama, Naoki Miyazaki.

**Funding acquisition:** Masayoshi Terayama.

**Methodology:** Toru Kitazawa.

**Project administration:** Masayuki Watanabe.

**Resources:** Kengo Kuriyama, Naoki Takahashi, Masahiro Tamura, Akihiko Okamura, Jun Kanamori.

**Supervision:** Toru Kitazawa, Misuzu Ishii, Kumi Takagi.

**Writing – original draft:** Masayoshi Terayama.

**Writing – review & editing:** Yu Imamura.

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
