## [Decision Letter · Decision Letter 0]

12 Mar 2025

PONE-D-24-44048
Study protocol: The effect of a low-carbohydrate enteral nutrition formula on postoperative hyperglycemia in non-diabetic patients with esophageal cancer: a randomized exploratory phase II trial (ENLICHE study)
PLOS ONE

Dear Dr. Imamura,

Thank you for submitting your manuscript to PLOS ONE. After careful consideration, we feel that it has merit but does not fully meet PLOS ONE’s publication criteria as it currently stands. Therefore, we invite you to submit a revised version of the manuscript that addresses the points raised during the review process.

We look forward to receiving your revised manuscript.

Kind regards,

Steven E. Wolf, MD

Academic Editor

PLOS ONE

Journal Requirements:

3. Please note that funding information should not appear in the Acknowledgments section or other areas of your manuscript. We will only publish funding information present in the Funding Statement section of the online submission form.  Please remove any funding-related text from the manuscript. 

Reviewers' comments:

Reviewer's Responses to Questions

**Comments to the Author**

1. Does the manuscript provide a valid rationale for the proposed study, with clearly identified and justified research questions?

Reviewer #1: Yes

Reviewer #2: Yes

2. Is the protocol technically sound and planned in a manner that will lead to a meaningful outcome and allow testing the stated hypotheses?

Reviewer #1: Yes

Reviewer #2: Yes

3. Is the methodology feasible and described in sufficient detail to allow the work to be replicable?

Reviewer #1: Yes

Reviewer #2: Yes

4. Have the authors described where all data underlying the findings will be made available when the study is complete?

Reviewer #1: Yes

Reviewer #2: Yes

5. Is the manuscript presented in an intelligible fashion and written in standard English?

Reviewer #1: Yes

Reviewer #2: Yes

6. Review Comments to the Author

You may also provide optional suggestions and comments to authors that they might find helpful in planning their study.

Reviewer #1: The authors present a study protocol to i investigate the impact of a low-carbohydrate EN

formula on hyperglycemic control during perioperative nutritional management of

esophageal cancer. They plan to recruit 50 patients to enroll in a single-center, randomized phase II trial with two arms (standard vs low-carbohydrate EN). Sample will be stratified by operation time and HbA1c. The primary outcome is the mean time-in-range across the 48 hours from postoperative day 1 to 2.

1. Line 6. Should more information, such as sex, diabetic status, etc, be collected at the enrollment as well?

2. Line 6. Please clarify when the enrollment for this study is, pre-operation or post-operation.

3. Line 18. The stratification is also based on HbA1c levels. Please clarify when this level is measured, enrollment, pre-operation, post-operation, or?

4. Line 23. “alternative formulations or reduce the dosage….” Will these participants be still included in the study? If so, how to deal with different treatments for different participants within the same group?

5. Line 15. “approximately 120 esophagectomies are performed annually.” Please clarify whether this refers to non-diabetic patients.

Reviewer #2: This study tackles an important but often overlooked issue in postoperative care—glycemic control in non-diabetic patients undergoing esophagectomy. The randomized controlled trial design gives it a solid foundation, but a few areas could be refined to improve clarity and real-world applicability. One key aspect that needs more attention is how preoperative metabolic variability and dietary habits might influence outcomes. Continuous glucose monitoring is a useful addition, but it would help to acknowledge its potential limitations, including sensor accuracy immediately after surgery and possible interference from perioperative medications. The stratification strategy based on operative time and HbA1c levels makes sense, but the decision to exclude pacemaker patients’ needs a clearer explanation. If it’s due to concerns about device interference or physiological differences, that should be explicitly stated.

This protocol has the potential to shape postoperative nutrition strategies for non-diabetic patients, an area where clear guidelines are still lacking. The manuscript makes a strong case for why glycemic management matters in this population, but widening the discussion to include other surgical contexts would make the findings even more relevant. While the hypothesis is well-reasoned, it would be helpful to consider real-world barriers to implementation, particularly in settings where specialized enteral formulas aren’t readily available.

Methodologically, the study is well-structured, with clear stratification criteria that improve comparability. The enteral nutrition protocol aligns with clinical practice, and safety measures for managing hypo- and hyperglycemia are well detailed. That said, the single-center design raises concerns about how generalizable the findings will be, and the open-label approach introduces potential bias. The authors should discuss how they plan to minimize these effects. The study also focuses narrowly on glucose control, but low carbohydrate enteral nutrition could have broader metabolic consequences, particularly on lipid metabolism and gut microbiota, which deserve at least a mention.

The statistical framework is solid, with time-in-range (TIR) as a meaningful primary outcome. The use of least squares mean difference and confidence intervals is appropriate, though it would be helpful to clarify how multiple comparisons are being handled. Given the likelihood of postoperative complications, a more detailed plan for managing missing data is needed. Subgroup analyses based on HbA1c levels or operative time could also add depth to the findings.

The manuscript is well-organized and follows transparency standards, with the SPIRIT checklist reinforcing protocol clarity. Minor refinements in the statistical methods section would improve readability. Ethical considerations are clearly addressed, with documented IRB approval and trial registration, which strengthens the study’s credibility.

While the authors acknowledge key limitations, they could go further in discussing long-term follow-up plans. The mention of a future phase III trial is promising, but it would be helpful to explain how this study’s results will shape that trial’s design. Addressing cost-effectiveness and practical barriers to using low carbohydrate enteral nutrition in routine care would also make the findings more applicable.

To strengthen this manuscript, the authors should clarify the rationale for excluding pacemaker patients, expand the discussion on metabolic effects beyond glucose control, provide a clearer strategy for handling missing data, and consider real-world implementation challenges. Refining statistical explanations and improving language precision would also enhance the manuscript.

Overall, this is a well-conceived and clinically relevant study. With a few revisions, it has strong potential for publication.

7. PLOS authors have the option to publish the peer review history of their article (what does this mean?). If published, this will include your full peer review and any attached files.

Reviewer #1: No

Reviewer #2: No

---

## [Author Response · Author response to Decision Letter 0]

21 Apr 2025

Terayama M et al. Manuscript ID “PONE-D-24-44048”:

Reply to Reviewers

To aid in the re-review of this manuscript, we have included a point by point response to each comment. Reviewers’ comments are shown in italics. We really appreciate the reviewers’ comments and suggestions. Because of these valuable comments, we believe that our manuscript has been much improved.

Reviewer #1: The authors present a study protocol to investigate the impact of a low-carbohydrate EN formula on hyperglycemic control during perioperative nutritional management of esophageal cancer. They plan to recruit 50 patients to enroll in a single-center, randomized phase II trial with two arms (standard vs low-carbohydrate EN). Sample will be stratified by operation time and HbA1c. The primary outcome is the mean time-in-range across the 48 hours from postoperative day 1 to 2.

1. Line 6. Should more information, such as sex, diabetic status, etc, be collected at the enrollment as well?

Response:

Thank you very much for your insightful comment. We would like to clarify that information such as sex, diabetic status, electrolyte status, and lipid status are already being collected as part of our study protocol. This information has been explicitly described in the Materials and Methods section as below:

“Registration and randomization

The following data will be preoperatively evaluated for study enrollment: age, sex, birth date, height, weight, blood pressure, heart rate, past medical history, comorbidities, allergies, oral medications, symptoms, Eastern Cooperative Oncology Group Performance Status (ECOG-PS), blood tests, including diabetic status, nutritional, lipid and electrolyte profiles, urine tests, electrocardiogram, chest X-ray, and computed tomography.”

 (Registration and randomization, in Materials and methods, page 6, lines 6-10)

2. Line 6. Please clarify when the enrollment for this study is, pre-operation or post-operation.

Response:

In this study, patients are registered preoperatively. For clarification, this information has been added to “Materials and Methods” section as below:

“Registration and randomization

The following data will be preoperatively evaluated for study enrollment: ….”

 (Registration and randomization, in Materials and methods, page 6, line 19)

3. Line 18. The stratification is also based on HbA1c levels. Please clarify when this level is measured, enrollment, pre-operation, post-operation, or?

Response:

In this study, the HbA1c levels are measured preoperatively. This information has been added to the Materials and methods section as below:

 “Stratification will also consider preoperative HbA1c levels (clinical pre-diabetic range of 6.0-6.4% vs. normal/near-normal range of <6.0%).”

 (Registration and randomization, in Materials and methods, page 6, line 19)

4. Line 23. “alternative formulations or reduce the dosage….” Will these participants be still included in the study? If so, how to deal with different treatments for different participants within the same group?

Response:

We appreciate the reviewer’s insightful comment. Patients who required alternative formulations or dose reductions were included in the Full Analysis Set, but excluded from the Per Protocol Set. The Full Analysis Set is used to reflect the actual clinical course of all enrolled cases, whereas the Per Protocol Set analysis is conducted to evaluate the pure effect of the originally assigned formulations in this study. This clarification has been added to the Materials and methods section as below:

“In the event that any cases require changes to the enteral nutrition formula or dosage reduction, a subgroup analysis will be conducted comparing groups that followed the protocol according to the original assignment (Per Protocol Set), excluding those cases. In principle, missing data will not be imputed. However, if necessary, sensitivity analyses incorporating imputation of missing data will be conducted. Specifically, if missing data occur during data collection, sensitivity analyses will be performed on cases in which more than 70% of the planned measurements have been obtained.”

(Statistical analysis, in Materials and methods, page 11, lines 3-9)

5. Line 15. “approximately 120 esophagectomies are performed annually.” Please clarify whether this refers to non-diabetic patients.

Response:

We appreciate the reviewer’s comment. In response to this comment, we have described the planned enrollment period based on the 120 cases experienced annually at our institution, taking into account the presence or absence of diabetes, in the Materials and methods section as below.

“Approximately 120 esophagectomies are performed each year at our institution. Among these, 12% (n = 14) are performed in patients with diabetes mellitus, indicating an estimated 106 non-diabetic cases annually, based on our previous study [13]. Given that approximately half of the non-diabetic patients are expected to be enrolled in this study, the enrollment of 50 cases is projected to be completed within 1.5 years.”

(Recruitment, in Materials and methods, page 10, lines 14-19)

Reviewer #2: This study tackles an important but often overlooked issue in postoperative care—glycemic control in non-diabetic patients undergoing esophagectomy. The randomized controlled trial design gives it a solid foundation, but a few areas could be refined to improve clarity and real-world applicability.

1. One key aspect that needs more attention is how preoperative metabolic variability and dietary habits might influence outcomes.

Response:

We appreciate the reviewer’s valuable comments. We agree that preoperative metabolic variability and dietary habits might affect postoperative glucose levels. We plan to comprehensively collect data on preoperative metabolic and nutritional status, to allow a subgroup analysis stratified by those metabolic status. We have revised the Material and Methods, and Discussion sections as below:

“Registration and randomization

The following data will be preoperatively evaluated for study enrollment: age, sex, birth date, height, weight, blood pressure, heart rate, past medical history, comorbidities, allergies, oral medications, symptoms, Eastern Cooperative Oncology Group Performance Status (ECOG-PS), blood tests, including diabetic status, nutritional, lipid and electrolyte profiles, urine tests, electrocardiogram, chest X-ray, and computed tomography.”

 (Registration and randomization, in Materials and methods, page 6, lines 6-10)

“Firstly, this study does not control potential influences of patients’ underlying metabolic disorders or dietary habits on postoperative glucose levels. However, in non-diabetic patients, there is no clear evidence indicating whether these factors significantly affect early postoperative glucose metabolism. In this study, we plan to conduct subgroup analyses based on metabolic and nutritional factors including triglycerides and LDL cholesterol to provide insights that may contribute to future research development.”

(Discussion, page 13, lines 14-19)

2. Continuous glucose monitoring is a useful addition, but it would help to acknowledge its potential limitations, including sensor accuracy immediately after surgery and possible interference from perioperative medications.

Response:

We appreciate the reviewer’s insightful comments. We have included those potential limitations regarding sensor accuracy immediately after surgery, and possible interference from perioperative medications, in the Material and Methods, and Discussion sections as below:

“Blood glucose levels will be continuously monitored from postoperative day (POD) 0 to POD 9. However, due to potential instability in measurements immediately following sensor application, data collected from POD 1 to POD 8 will be used for analysis.”

(Glucose monitoring in Material and Methods, page 8, lines 13-16)

“Secondly, there is a potential limitation associated with instability in glucose measurements during the initial 8 to 12 hours following sensor insertion, which is performed in the immediate postoperative period. To minimize such variability, data collection begins on the day after surgery, allowing at least 12 hours to elapse after sensor placement. Thirdly, the potential influence of differences in perioperative medications on glucose measurements cannot be completely eliminated. However, this study includes only non-diabetic patients, and individuals taking oral medications that directly affect blood glucose levels are excluded. In our clinical practice, medications not essential for life support—such as antiplatelet and antiepileptic agents—are generally resumed from postoperative day 3. Moreover, any newly initiated medications in the postoperative period are standardized across all patients. Therefore, we believe that the primary endpoint, assessed based on blood glucose levels up to postoperative day 2, is unlikely to be significantly affected.”

(Discussion, page 13 line 20 – page14 line 8)

3. The stratification strategy based on operative time and HbA1c levels makes sense, but the decision to exclude pacemaker patients’ needs a clearer explanation. If it’s due to concerns about device interference or physiological differences, that should be explicitly stated.

Response:

We appreciate the reviewer’s comments. The reason for excluding pacemaker patients is stated in the Material and Methods section as below:

“4) have an implanted pacemaker with a reported potential for causing malfunctions in blood glucose sensors.”

(Exclusion Criteria, in Material and Methods, page 6, line 3)

4. This protocol has the potential to shape postoperative nutrition strategies for non-diabetic patients, an area where clear guidelines are still lacking. The manuscript makes a strong case for why glycemic management matters in this population, but widening the discussion to include other surgical contexts would make the findings even more relevant. While the hypothesis is well-reasoned, it would be helpful to consider real-world barriers to implementation, particularly in settings where specialized enteral formulas aren’t readily available.

Response:

The reviewer has raised an extremely important point. The potential effect of a low-carbohydrate enteral formula in suppressing postoperative hyperglycemia in non-diabetic patients is being investigated for the first time in this study. If such an effect is demonstrated, we believe that the medical environment should be adapted to allow for more widespread use of low-carbohydrate enteral nutrition. It would be challenging to achieve effective carbohydrate restrictions during the early postoperative periods by using alternative ways. We have revised the manuscript, accordingly, taking this perspective into account.

“Finally, as this is a pilot study with a small sample size conducted at a single institution, the generalizability of the findings may be limited. …. In addition, a reduction in infectious complications would demonstrate the cost-effectiveness of low-carbohydrate enteral nutrition, supporting its broader adoption in perioperative management—not only for non-diabetic patients undergoing esophagectomy, but also across a wide range of surgical procedures. Given that the cost of low-carbohydrate enteral nutrition is not significantly different from that of conventional formulas, we believe that, if this study confirms its effectiveness in suppressing early postoperative hyperglycemia, the medical environment should be adapted to facilitate its wider implementation.”

 (Discussion, page 14, lines 8-22)

5. Methodologically, the study is well-structured, with clear stratification criteria that improve comparability. The enteral nutrition protocol aligns with clinical practice, and safety measures for managing hypo- and hyperglycemia are well detailed. That said, the single-center design raises concerns about how generalizable the findings will be, and the open-label approach introduces potential bias. The authors should discuss how they plan to minimize these effects. The study also focuses narrowly on glucose control, but low carbohydrate enteral nutrition could have broader metabolic consequences, particularly on lipid metabolism and gut microbiota, which deserve at least a mention.

Response:

We appreciate the reviewer’s important comment. We agree that a single-center design may limit the generalizability of our findings and that an open-label approach can introduce bias. To overcome these essential issues, a future multi-center, double-blind phase III trial is required to validate and extend our findings across diverse populations. These considerations have been added to the Discussion section accordingly. Furthermore, we have also included a statement in the Discussion regarding the potential effects of low-carbohydrate enteral nutrition on lipid metabolism and gut microbiota, as below:

“Finally, as this is a pilot study with a small sample size conducted at a single institution, the generalizability of the findings may be limited. In addition, the open-label design may introduce potential bias. If the efficacy and safety of a low-carbohydrate nutritional formula in non-diabetic patients are confirmed, the findings will serve as confirmatory evidence to justify a double-blind, multi-center phase III trial aimed at further evaluating the impact of low-carbohydrate enteral nutrition on reducing PICs after esophagectomy. Furthermore, such a trial should also investigate the potential effects of low-carbohydrate enteral nutrition on lipid metabolism and gut microbiota.”

(Discussion, page 14, lines 8-15)

6. The statistical framework is solid, with time-in-range (TIR) as a meaningful primary outcome. The use of least squares mean difference and confidence intervals is appropriate, though it would be helpful to clarify how multiple comparisons are being handled. Given the likelihood of postoperative complications, a more detailed plan for managing missing data is needed. Subgroup analyses based on HbA1c levels or operative time could also add depth to the findings. The manuscript is well-organized and follows transparency standards, with the SPIRIT checklist reinforcing protocol clarity. Minor refinements in the statistical methods section would improve readability. Ethical considerations are clearly addressed, with documented IRB approval and trial registration, which strengthens the study’s credibility.

Response:

We appreciate the reviewer’s insightful comments.

In this study, we have defined a single primary outcome, while multiple outcomes will be analyzed as secondary outcomes. Because this is an exploratory study with a limited sample size, we do not plan to adjust for multiplicity in the analysis of secondary outcomes. Instead, adjustments for multiplicity will be considered in the design of a future phase III trial.

In principle, missing data will not be imputed. However, if necessary, sensitivity analyses incorporating imputation of missing data will be conducted. Specifically, if missing data occur during data collection, sensitivity analyses will be performed on cases in which more than 70% of the planned measurements have been obtained. This description has been added to “Discussion” section as below:

“Given the exploratory nature of this study and the limited sample size, adjustments for multiplicity in the analysis of secondary outcomes will not be performed.”

(Statistical analysis, in Material and Methods, page 11, lines 15-17)

“In principle, missing data will not be imputed. However, if necessary, sensitivity analyses incorporating imputation of missing data will be conducted. Specifically, if missing data occur during data collection, sensitivity analyses will be performed on cases in which more than 70% of the planned measurements have been obtained.”

(Discussion, page 11, line 6-9)

7. While the authors acknowledge key limitations, they c

---

## [Decision Letter · Decision Letter 1]

7 May 2025

Study protocol: The effect of a low-carbohydrate enteral nutrition formula on postoperative hyperglycemia in non-diabetic patients with esophageal cancer: a randomized exploratory phase II trial (ENLICHE study)

PONE-D-24-44048R1

Dear Dr. Imamura,

We’re pleased to inform you that your manuscript has been judged scientifically suitable for publication and will be formally accepted for publication once it meets all outstanding technical requirements.

Kind regards,

Steven E. Wolf, MD

Academic Editor

PLOS ONE

Additional Editor Comments (optional):

Reviewers' comments:

Reviewer's Responses to Questions

**Comments to the Author**

1. Does the manuscript provide a valid rationale for the proposed study, with clearly identified and justified research questions?

Reviewer #1: Yes

Reviewer #2: Yes

2. Is the protocol technically sound and planned in a manner that will lead to a meaningful outcome and allow testing the stated hypotheses?

Reviewer #1: Yes

Reviewer #2: Yes

3. Is the methodology feasible and described in sufficient detail to allow the work to be replicable?

Reviewer #1: Yes

Reviewer #2: Yes

4. Have the authors described where all data underlying the findings will be made available when the study is complete?

Reviewer #1: Yes

Reviewer #2: Yes

5. Is the manuscript presented in an intelligible fashion and written in standard English?

Reviewer #1: Yes

Reviewer #2: Yes

6. Review Comments to the Author

You may also provide optional suggestions and comments to authors that they might find helpful in planning their study.

Reviewer #1: Thanks for your appropriately addressing all the raised comments and concerns with updated manuscript. This reviewer has no further concerns.

Reviewer #2: The authors have satisfactorily addressed all of my prior concerns and clarified all points raised in my review

7. PLOS authors have the option to publish the peer review history of their article (what does this mean?). If published, this will include your full peer review and any attached files.

Reviewer #1: No

Reviewer #2: No

---

## [Editor Report · Acceptance letter]

PONE-D-24-44048R1

PLOS ONE

Dear Dr. Imamura,

I'm pleased to inform you that your manuscript has been deemed suitable for publication in PLOS ONE. Congratulations! Your manuscript is now being handed over to our production team.

Kind regards,

on behalf of

Dr. Steven E. Wolf

Academic Editor

PLOS ONE